# A Bulk Acoustic Wave Strain Sensor for Near-Field Passive Wireless Sensing

**DOI:** 10.3390/s23083904

**Published:** 2023-04-12

**Authors:** Xiyue Zou, Li Wen, Bin Hu

**Affiliations:** 1Key Laboratory of Nondestructive Testing and Evaluation for State Market Regulation, China Special Equipment Inspection and Research Institute, Beijing 100029, China; 2Department of Mechanical Engineering and Automation, Beihang University, Beijing 100191, China

**Keywords:** bulk acoustic wave device, passive wireless sensor, strain measurement, structural health monitoring, wireless power transfer

## Abstract

Near-field passive wireless sensors can realize non-contact strain measurement, so these sensors have extensive applications in structural health monitoring. However, these sensors suffer from low stability and short wireless sensing distance. This paper presents a bulk acoustic wave (BAW) passive wireless strain sensor, which consists of two coils and a BAW sensor. The force-sensitive element is a quartz wafer with a high quality factor, which is embedded into the sensor housing, so the sensor can convert the strain of the measured surface into the shift of resonant frequency. A double-mass-spring-damper model is developed to analyze the interaction between the quartz and the sensor housing. A lumped parameter model is established to investigate the influence of the contact force on the sensor signal. Experiments show that a prototype BAW passive wireless sensor has a sensitivity of 4 Hz/με when the wireless sensing distance is 10 cm. The resonant frequency of the sensor is almost independent of the coupling coefficient, which indicates that the sensor can reduce the measurement error caused by misalignment or relative movement between coils. Thanks to the high stability and modest sensing distance, this sensor may be compatible with a UAV-based monitoring platform for the strain monitoring of large buildings.

## 1. Introduction

In structural health monitoring (SHM), strain sensors are widely used to inspect local stresses in large structures. However, conventional strain gauges suffer from wire redundancy. Although active wireless strain sensors composed of strain gauges and wireless nodes solve the problem of wiring, these wireless sensors need to replace batteries periodically. As an alternative solution, passive wireless strain sensors, which are resonator-based, battery-free, and non-tether electronic devices, can perform non-contact strain measurements. Researchers have investigated several types of passive wireless sensors, which can be classified as near-field and far-field passive wireless sensors based on their wireless power transfer method [1]. Far-field passive wireless sensors use dipole or patch antennas to receive and reflect electromagnetic waves in the ultrahigh frequency band; the sensing distance of these sensors is usually greater than 0.5 m [2,3,4,5,6,7,8]. However, their operating frequency overlaps with that of wireless communications, such as TVs, FM radios, and mobile phones, so the signal of sensors could be affected by these interfering sources, which brings difficulties to signal processing [9]. In contrast, near-field passive wireless sensors operate in the middle or high-frequency band, using two coils to transmit energy at a short distance. The most common near-field passive wireless sensor is based on the inductor-capacitor (LC) circuit, including a capacitive sensor and an inductive coil [10,11,12]. Unfortunately, their wireless sensing distance is relatively low, due to the low quality factor and the low transmitter gain. Therefore, this problem limits the application of near-field passive wireless sensors.

In this paper, we present a novel near-field passive wireless strain sensor based on a bulk acoustic wave (BAW) device. Its sensing element is a quartz resonator sandwiched in the sensor housing. The high quality factor of the resonator improves the efficiency of the wireless power transmission and the signal-to-noise ratio. Current BAW sensors, such as quartz crystal microbalances (QCMs), have been widely used in chemical and biological analytics [13], temperature and humidity monitoring [14], and pressure detection [15]. Previous efforts have confirmed that a quartz wafer can be simplified as a mass-spring-damper model [16,17]. For biochemical sensors, when liquid or gas flows through the wafer, the mass of the coating on the wafer surface will increase in response to the analyte in the flow, so the “added mass” will reduce its resonant frequency [18,19]. For pressure sensors, when a fluid passes through the wafer, the wafer is subjected to pressure from the fluid flow, so the equivalent “added stiffness” will increase its resonant frequency [15]. To the best of our knowledge, no bulk acoustic wave force/strain sensor has been proposed in the literature, but previous efforts have investigated the contact mechanism between a quartz wafer and a contact element [20,21]. The principle is similar to BAW pressure sensors. The contact forces can be considered as an “added stiffness” of the resonator, so the resonant frequency increases with normal force.

In this paper, we present a BAW passive wireless strain sensor. In Section 2, we develop an analytical model to investigate the principle of a BAW strain sensor. Then, we briefly describe the analytical model of a BAW passive wireless sensing sensor. In Section 3, we characterize the relationship between the normal force and the resonance shift of the quartz. A numerical model illustrates the availability of the sensor design. Finally, we use experimental demonstrations to study the performance of a prototype sensor, such as sensitivity and sensing distance.

## 2. Materials and Methods

In this paper, a BAW strain sensor makes use of four rules. The schematic is shown in Figure 1.

First, a BAW passive wireless strain sensor consists of two subsystems: a transmitter and a receiver. The receiver has two components: an inductive coil and a BAW strain sensor. As a force-sensitive element, a quartz wafer is embedded in the sensor housing. When the excitation frequency is equal to the resonant frequency of the receiver, the impedance of the receiver reaches a global minimum value, whereas that of the transmitter reaches a local maximum value. Therefore, we can detect the peak signal of the transmitter to analyze the deformation of the substrate of the BAW sensor.

Second, the BAW strain sensor has a sandwich-like structure, which converts the deformation of the measured structure into the resonance shift of a quartz resonator. A quartz wafer with a thickness shear vibration mode generates bulk transverse waves that travel normally to the plate surface, so the wafer surface moves parallel to the surface. If an object contacts the surface of the wafer, the wafer is subjected to compression and friction from the object. If we consider the quartz wafer as a mass-spring-damper resonator, these external forces can be considered as an “added stiffness”. Previous literature has reported that the resonance shift is proportional to the contact radius between the quartz plate and the contact unit [20]. Based on the contact mechanism of piezoelectric materials, we designed a sensor housing, which converts the stress of the substrate into the contact forces applied to the upper surface of the wafer. The bottom of the sensor housing is in contact with two boundaries of the wafer’s lower surface, so the air gap allows the wafer to vibrate.

Third, we chose a thin ceramic sheet as the contact unit to improve the sensitivity of the sensor. The objective of the sensor design was to continuously change the contact area between the contact unit and the quartz wafer, so the contact unit should have a high stiffness. Otherwise, a soft contact unit, such as silicon rubber, would fully adhere to the wafer even if the contact force is very low, so the resonance will not change with the contact force. Furthermore, we chose a flat contact surface, instead of a spherical surface, to enhance the sensitivity of the sensor. The experimental studies in previous work have shown that the resonance shift was below 200 Hz when a spherical ball made of ceramic contacts a quartz wafer with a resonance of 10–12 MHz [21]. The low sensitivity is due to a nonlinear relationship between the contact radius and normal force. If the contact surface is flat, the contact area will linearly increase with the normal force due to the non-uniform contact pressure.

Fourth, the contact unit was connected with the sensor housing via a deformable force buffering structure, which protects the vulnerable quartz wafer from the damage of overlarge contact pressure. In this paper, a soft cushion made of rubber was used to demonstrate the feasibility of the sensor, but the buffering structure can be a spring or a thin-walled deformable structure.

The primary design parameter of a BAW passive wireless sensor is the resonant frequency of the quartz wafer. A thinner quartz wafer has a higher resonant frequency and a lower resistance. To decrease the resistance of the receiver, we prefer to choose a wafer with a higher resonance frequency. However, the strength of a quartz wafer decreases with its thickness so that a thin wafer is unable to withstand the applied normal force. Due to the restricted manufacturing conditions, we composed the parts of the sensor manually, so the highest resonant frequency of the sensor we could fabricate was 10 MHz; the thickness of the quartz wafer was approximately 0.16 mm, and the equivalent resistance was near 37 Ohms. Secondary design parameters include the impedance of the inductive coil and the gain of the transmitter. This paper does not consider the coil optimization, but we use an RLC circuit to compensate for the impedance of the transmitter.

In this section, we establish an analytical model to investigate the properties of a quartz resonator in contact with an object. The governing equations of a quartz wafer are represented by the following equations [22]:(1)$$T_{ij}=C_{ijkl}E_{kl}+\eta_{ijkl}{\dot{E}}_{kl}+e_{pij}\phi_{,p}$$
(2)$$D_{i}=e_{iq}E_{q}+\varepsilon_{ik}\phi_{,k}$$
where *T* is stress, *D* is electrical displacement, *E* is strain, *ϕ* is electrical potential; *C*, *η*, *e*, and *ε* are the coefficient of stiffness, damping, piezoelectric, and dielectric, respectively. The resonant frequency of a piezoelectric plate with infinite length can be written as [20]:(3)$$f_{s}=\frac{1}{2t}\sqrt{\frac{c_{66}}{\rho }}$$
where *t* is thickness, *ρ* is density, and $c_{66}$ is a coefficient in the stiffness matrix. The equation suggests that the resonance of a thickness shear mode piezoelectric plate mainly depends on its shear stiffness, so we ignore the normal contact stiffness *k_n_* in this paper.

We establish a dual mass-spring-damper system to analyze the influence of the contact unit on the wafer (see Figure 2a).

Assuming each object is a square plate, the effective mass can be written as [15]:(4)$$m_{i}=\rho_{i}h_{i}L_{i}^{2}$$
where *L* and *h* are the length and height of the object, respectively; subscripts 1 and 2 represent the quartz resonator and the contact unit, respectively. The resonant frequency of a single mass-spring-damper model can be represented as:(5)$$f_{s,i}=\frac{1}{2\pi }\sqrt{\frac{k_{i}}{m_{i}}}$$
where *k* is the effective stiffness. According to Equations (3)–(5), we can find the effective stiffness of a quartz oscillator as follows:(6)$$k_{i}=\frac{\pi^{2}G_{i}L_{i}^{2}}{h_{i}}$$
where *G*_1_ is the coefficient *c_66_* of the quartz wafer; *G*_2_ is the shear modulus of the contact unit.

The two objects are coupled by a spring *k_c_* and a damper *b_c_*. The equations of motion of the system can be written as:(7)$$\left[\begin{array}{cc} m_{1} & 0\\ 0 & m_{2} \end{array}\right]\left[\begin{array}{c} {\ddot{x}}_{1}\\ {\ddot{x}}_{2} \end{array}\right]+\left[\begin{array}{cc} b_{1}+b_{c} & -b_{c}\\ -b_{c} & b_{2}+b_{c} \end{array}\right]\left[\begin{array}{c} {\dot{x}}_{1}\\ {\dot{x}}_{2} \end{array}\right]+\left[\begin{array}{cc} k_{1}+k_{c} & -k_{c}\\ -k_{c} & k_{2}+k_{c} \end{array}\right]\left[\begin{array}{c} x_{1}\\ x_{2} \end{array}\right]=\left[\begin{array}{c} 0\\ 0 \end{array}\right]$$
(8)$$m_{1}{\ddot{x}}_{1}+b_{1}{\dot{x}}_{1}+k_{1}x_{1}=-k_{c}(x_{1}-x_{2})-b_{c}({\dot{x}}_{1}-{\dot{x}}_{2})$$
where the first term on the right side of Equation (8) depends on the contact forces, and the second term depends on the energy loss of the wafer, including acoustic wave emissions and heat loss. To study the influence of contact forces on the resonance shift of the quartz, we ignore the damping coefficients, so Equation (7) can be written as follows:(9)$$m_{1}{\ddot{x}}_{1}+(k_{1}+k_{c})x_{1}-k_{c}x_{2}=0$$
(10)$$m_{2}{\ddot{x}}_{2}+(k_{c}+k_{2})x_{2}-k_{c}x_{1}=0$$

The eigenfunction and eigenvalues of Equations (9) and (10) are determined by the following equations:(11)$$m_{1}m_{2}\lambda^{2}-\left(m_{1}\left(k_{2}+k_{c}\right)+m_{2}\left(k_{1}+k_{c}\right)\right)\lambda +\left(k_{1}k_{2}+\left(k_{1}+k_{2}\right)k_{c}\right)=0$$
(12)$$\lambda_{1,2}=\frac{1}{2m_{1}m_{2}}\left[m_{1}\left(k_{2}+k_{c}\right)+m_{2}\left(k_{1}+k_{c}\right)\pm \sqrt{m_{1}^{2}{\left(k_{2}+k_{c}\right)}^{2}+m_{1}^{2}{\left(k_{1}+k_{c}\right)}^{2}-2m_{1}m_{2}\left(k_{1}k_{2}+\left(k_{1}+k_{2}\right)k_{c}\right)}\right]$$

If the contact surface is flat and parallel to the quartz wafer, the contact stiffness in the tangential direction can be represented as the following equations [21]:(13)$$k_{c,\max}=k_{t}=2G^{*}a$$
(14)$$\frac{1}{G^{*}}=\frac{2-\nu_{1}}{4G_{1}}+\frac{2-\nu_{2}}{4G_{2}}$$
where *G** is the effective shear modulus. Due to the non-uniform pressure distribution on the contact surface, the contact stiffness will increase with the contact forces, so the tangential contact stiffness *k_t_* is the maximum value of contact stiffness. For the real sensor, the parallelism error between the wafer and the ceramic sheet will affect the sensitivity of the sensor as well.

In Case A, we substitute each coefficient in the BAW sensor into Equation (12), see Table 1. The relationship between *k_c_* and *λ*_1_ is shown in Figure 2b. According to Equation (13), the maximum contact stiffness *k_c_*_,max_ is 5.52 × 10^8^ N/m. The expression of eigenvalues is highly non-linear, so the resonant frequency increases nonlinearly with the contact stiffness. If the contact stiffness could reach its maximum value, the resonance shift is 13.2 kHz, and the eigenvalue is ***u***_1_ = [−1.08, 0.0028]^T^. The result suggests that the contact has little influence on the thickness-shear vibration mode of the oscillator.

In Case B, the mass and stiffness of a soft contact unit have little influence on the resonance when the contact stiffness is the same. However, the maximum contact stiffness is limited by the stiffness and dimensions of the contact unit, so the resonance will be constant after the soft material fully adheres to the wafer.

## 3. Results

### 3.1. Experimental Characterization of Quartz Resonator with Contact Force

In order to prove the validity of the contact model, we use an experimental method to characterize the relationship between the resonant frequency of a quartz wafer and a payload applied on its top surface. Due to the lack of a wafer probe station, we built a simple loading platform to apply weights to the wafer in the vertical direction. We removed the shell of commercial crystal oscillators to obtain quartz wafers with a resonance of 10 MHz. Then we applied different weights on the top of the loading cell, including 2 g, 5 g, 10 g, 20 g, and 50 g. When the contact unit is made of ceramic and rubber, the variation of the conductance of the quartz resonator is shown in Figure 3a,b.

The first contact unit is a ceramic sheet with a dimension of 2 mm × 2 mm × 0.2 mm. Figure 3a shows that the resonant frequency of the quartz wafer increases nonlinearly with the payload due to the nonlinear relationship between the contact stiffness and the resonance shift. Furthermore, the magnitude of conductance decreases with the payload due to the energy loss. When the payload is 50 g, the resonance shift is 6.08 kHz, which is in the same magnitude as the analytical solutions. By comparing with the theoretical maximum resonance shift, 13.2 kHz, we can calculate that the actual contact radius is 46% of the maximum contact radius. When the payload is over 100 g, the contact unit would break the wafer. The second contact unit is a rubber block with a dimension of 5 mm × 5 mm × 2 mm. The resonance is approximately constant because of its low stiffness. These results are consistent with the results in the contact model.

In the next step, we extract the effective contact spring *k_c_* and the damping coefficient *b_c_* in the analytical model from the experimental results. In the analytical model, we input a unit step displacement to Object 1 (the quartz) and calculate the displacement of each object by Equation (7). First, we determine the contact spring coefficient. We set all damping coefficients to 0 and adjust the value of *k_c_* until the resonance shift matches the corresponding experimental results. Second, we determine the contact damping coefficient. We assume that the damping coefficient does not affect the resonant frequency because the damping coefficient is relatively low. In the frequency domain, we can calculate a quality factor according to the bandwidth of measured conductance. In the time domain, we optimize the damping coefficient in the analytical model until the amplitude decay in the analytical model matches the quality factor in the experiment. Figure 3c shows that both *k_c_* and *b_c_* increase nonlinearly with the payloads. In order to validate the contact stiffness, we substitute *k_c_* into Equation (12), using the contact stiffness to calculate the resonance shift and comparing it with that in the experiment. The results in Figure 3d show that the analytical solutions match the experimental data.

To create a numerical model of the BAW sensor, we start with the model of the force-sensing element. According to the geometric symmetry, we simplify the model into a two-dimensional model to speed up the calculation. The model includes two steps: a static analysis and a harmonic analysis. In the first step, we calculate the stress distribution of the wafer when a payload is applied to the contact unit. In the second step, we use the solution in the first step as initial conditions to calculate the frequency response of the wafer. More details of the model are shown in Appendix B. Figure 4a shows the boundary conditions of the model and the vibration of the wafer at the resonant frequency. A force is applied to the center of the contact unit in the direction of -y. Figure 4b shows the influence of contact forces on the conductance of a wafer when different payloads were applied to the contact unit. As a validation, we compare the resonance of the wafer in the numerical model with that in the experiment. The numerical results basically match the experimental results, whereas the sensitivity in the experiment is slightly lower than that in the simulation. It may be due to the error of material parameters, such as damping coefficient and stiffness. In addition, when the contact force is the same, the simplification of the model increases the changes in the contact area.

### 3.2. Design of a BAW Strain Sensor

In this subsection, we present the design of a prototype BAW strain sensor (see Figure 5a). The substrate is made of 3003 aluminum alloy foil with a thickness of 0.2 mm. A quartz oscillator with a thickness of 0.16 mm is supported by two rectangular blocks made of Alumina ceramic, called left and right supports. We used adhesive (LOCTITE 403) to fix one wire lead of the resonator to the left support. This quartz wafer is too vulnerable to withstand the pressure from a deformed rigid structure, so we use a soft spacer made of silicon rubber to decrease the force applied to its upper surface. To improve the compliance of the sensing housing, we use a bridge-shaped aluminum sheet to connect the left and right support. When the bridge is fixed to the two supports, the contact unit can apply a preload to the upper surface of the wafer. This sensor can detect the bending of the attached structure. When the measured structure is bent into an upper convex shape, the angle between the two supports will change. Thus, the bridge and soft spacer will move downward and compress the wafer.

To verify the feasibility of the design, we establish a numerical model of the sensor in COMSOL 5.5; more details about the model are shown in Appendix B. We apply a downward force to the free end of a cantilever beam, so the strain of the substrate increases with the payload. Figure 5b shows the deformation of the sensor housing when the beam is bent to different curvatures. The ratio of the maximum stress on the wafer to the maximum stress on the surface of the beam is approximately 13%, which confirms that the sensor housing decreases the pressure applied to the wafer. Figure 5c shows that the contact force increases linearly with the strain; when the strain of the measured surface is approximately 800 με, the contact force is 0.63 N, which is lower than the maximum allowed payloads. As shown in Figure 5d, the distribution of stress on the contact surface also suggests that the contact radius increases with the measured strain.

The performance of the sensor can be affected by manufacturing errors. The preload applied to the wafer is one of the most influencing factors. In the case study, we fabricated two sensors with different preloads (see Figure 6). When we glued the bridge on two supports, we applied a lower pressure to the bridge of Sensor A, compared to Sensor B. As a result, Sensor A has a lower sensitivity and a higher measurement range than Sensor B, and they have different response curves. In Figure 6a, the calibration curve includes three stages: the initial stage, the operation stage, and the final stage. In the initial stage, the contact unit slightly touches the wafer. Due to the nonlinear relationship between contact stiffness and resonance shift, the sensitivity in this stage is much lower than that in the next stage. In the operation stage, both sensitivity and linearity are improved because the contact force increases. In the final stage, the sensitivity is reduced due to the contact area approaching its maximum. In Figure 6b, the calibration curve skipped the initial stage due to a higher initial contact force. Therefore, the maximum strain that the sensor can measure decreases with increasing preloads. Besides the preload applied to the wafer, the position error and parallelism errors between the contact unit and the wafer affect the sensor performance as well. For example, the maximum resonance shift of Sensor A is lower than that of Sensor B; this may be due to the position error or parallelism errors between the contact unit and the wafer. The bulk transverse waves only generate in electrodes; if there is an offset between the contact area and the center of the electrode, the sensitivity will be reduced. Similarly, the parallelism error decreases the contact area, which will reduce the maximum resonance shift.

### 3.3. Characterization of a BAW Passive Wireless Sensor

In this subsection, we establish a lumped circuit model to study how the contact force affects the signal of the passive wireless sensor. The equivalent electrical circuits are shown in Figure 7a. The circuit model of a quartz oscillator is determined by four lumped parameters. Its impedance can be represented as:(15)$$Z_{PZT}=\frac{\left(R_{1,pzt}+j\omega L_{1,pzt}+{\left(j\omega C_{1,pzt}\right)}^{-1}\right)\cdot {\left(j\omega C_{0,pzt}\right)}^{-1}}{R_{1,pzt}+j\omega L_{1,pzt}+{\left(j\omega C_{1,pzt}\right)}^{-1}+{\left(j\omega C_{0,pzt}\right)}^{-1}}$$
where *C*_0,*pzt*_ is the capacitance of electrodes; *L*_1,*pzt*_, and *C*_1,*pzt*_ are determined by the resonant frequency of the wafer; *R*_1,*pzt*_ is determined by its energy loss. We measure the impedance of a quartz wafer and extract these four parameters. Table 2 shows the effect of the payload on the effective lumped parameters.

The effect of the contact force on each circuit element can be considered as an added element, which is connected in series with the original element (see Figure 7b). The parameters *R*_1_ and *L*_1_ increase linearly with the payload because the energy loss and contact force of the wafer are linearly related to the payload. The capacitance parameters *C*_1,*c*_ and *C*_0,*c*_ decrease nonlinearly with the payload, which is determined by the variation of the resonant and anti-resonant frequencies. Each capacitance parameter can be approximately fitted to a power function *y* = *kx*^−1^.

The impedance of the receiver and the transmitter can be written as:(16)$$Z_{r}=Z_{L,r}+Z_{PZT}$$
(17)$$Z_{t}=Z_{L,t}+Z_{C,t}+\frac{{\left(\omega M\right)}^{2}}{Z_{r}}$$
(18)$$S_{11}{(\mathrm{dB})=20\log}_{10}\left(\left|\frac{Z_{out}-Z_{in}}{Z_{out}+Z_{in}}\right|\right)$$
where the impedance of the coil can be written as *Z_L_*_,*x*_ =*R_x_* + *jωL_x_*, the impedance of tuning capacitor can be written as *Z_C_*_,*t*_ = (*jωC_t_*)^−1^, and the mutual inductance can be written as *M* = *κ*(*L_r_L_t_*)^1/2^.

In the analytical model, all lumped parameters are determined by the measurement results, see Table 3. We use two commercial inductive coils to test the prototype passive wireless sensor. Due to the parasite capacitance of coils, the capacitance of the tuning capacitor *C_t_* in the model is determined by the equation *f_s_* = (2*π*)^−1^(*C_t_L_t_*)^−1/2^, where *f_s_* is the resonance of the receiver.

First of all, let us assume that no force is applied to the resonator. The analytical results of the signals are shown in Figure 8. When two subsystems are decoupled, the reflection of the transmitter is an individual peak signal with a wide bandwidth and a low quality factor of 12. As shown in Figure 8a, when *κ* increases from 0 to 0.2, the reflection becomes an overlapping peak signal at the resonant frequency of the receiver, which includes two individual peak signals of the transmitter and the receiver; the bandwidth of the latter is much lower than that of the former. When *κ* increases from 0 to 0.05, the resonance of the transmitter decreases, whereas the resonance of the receiver is almost constant. Previous efforts illustrate that the resonance shift of a coupled system depends on both its energy loss rate and coupling coefficient [22,23]. In our methods, the quality factor of the receiver is much higher than that of the transmitter. Thus, if *κ* is relatively low, the variation of *κ* has little influence on the resonant frequency. Therefore, when demodulating signals, we extract only the resonant frequency of the receiver’s peak signal to calculate the strain without using the magnitude of the reflection at this frequency. This is because the resonant frequency is only determined by the contact forces on the resonator, whereas the magnitude of the reflection at this frequency is determined by both the strain and the coupling coefficient. If the gap between coils has a small offset, the magnitude of the reflection will change significantly, thus causing inconvenience to the signal processing. Although we do not use the magnitude of the reflection to calculate the strain, a sharp reflection curve can improve the resolution of the sensor. When *κ* increased from 0 to 0.038, the resistance of the transmitter increased from 33.5 Ohms to approximately 50 Ohms, so the reflection reaches a maximum value; this gap between coils can be considered as an optimal position.

Then, we study the effect of payloads on the signal. Assuming that *κ* is 0.05, we substitute the effective lumped parameters in Table 2 into Equations (16)–(18). Figure 8b shows that the frequency of the receiver’s peak signal increases with the payloads. When the force increases to 0.02 N, the magnitude of the peak signal is significantly improved because the impedance of the transmitter decreases to approximately 50 Ohms. The results suggest that the optimal position of the passive wireless sensor is tunable via tuning *R_t_*.

Based on the above analytical results, we investigate the influence of *R_t_* on the signal. Assuming that no force is applied to the wafer when *R_t_* is 50 Ohms, the optimal position is at infinity (*κ* = 0) (see Figure 8c). If the gap between coils decreases (e.g., *κ* = 0.01), the intensity of the receiver’s signal will be significantly greater than that when *R_t_* is 33.5 Ohms. Due to the errors of resistance and capacitors, it is difficult to perfectly realize the above results in experiments, whereas the results suggest two conclusions. First, we can adjust the resistor of the transmitter *R_t_* to enhance the resolution of the sensor. Second, the optimal position between coils increases with increasing *R_t_*, so we can adjust the optimal position of the sensor. Like the impedance matching of the antenna, this method maximizes the output power of the transmitter; more importantly, this method increases the magnitude and sharpness of the receiver’s signal such that we can find the resonant frequency more accurately. This method may be useful for specific applications where the gap between coils is constant, such as a receiver coil embedded in a composite material.

A simple demonstration is shown in Figure 8d. We compensated the imaginary part of the transmitter by tuning the capacitors. In the next step, we connected a few of the resistors in series with tuning capacitors to increase the real part of the transmitter to approximately 50 Ohms. In the demonstration, the gap between coils was 10 cm. Due to the resistance of wirings, the magnitude of the reflection of the transmitter was lower than what we expected, but the magnitude of the receiver was improved from 0.023 to 0.1. In conclusion, the compensation of the real part of the transmitter can improve the magnitude of the receiver’s signal, thus improving the resolution of the passive wireless sensor.

### 3.4. Experimental Demonstration of a BAW Passive Wireless Sensor

In this section, we demonstrate the performance of a prototype BAW passive wireless strain sensor. Due to the restrictions of experimental conditions, we only verified the sensor in the laboratory. The reflection of the transmitter was measured by a Keysight E5061b ENA vector network analyzer. The room temperature and humidity were approximately 22 °C and 33%, respectively. A prototype BAW strain sensor was attached to a cantilever beam (see Figure 9a); each kilogram of weight applies a deformation of 100 με to the beam. The tuning circuits consist of two capacitors connected in parallel. We tuned the value of each capacitor so that the resonance of the transmitter was approximately equal to that of the receiver. The receiver coil and the transmitter coil (6.3 μH and 43 mm in diameter) are connected to the sensor and the tuning circuit, respectively. To adjust the gap between the two coils, we attached each coil to the side of an individual test rig. As shown in Figure A2 in Appendix C, we used a numerical method to determine the relationship between the coupling coefficient and the separation distance between coils. According to this relationship, we can compare the experimental results with the analytical results to verify the analytical model of the BAW passive wireless sensor. Figure A3 in Appendix C shows that the experimental results match the analytical results when the gap is different.

The first experiment proves that the passive wireless sensor has an optimal position, where the resolution is maximized. The analytical results in Figure 8a suggest that the corresponding coupling coefficient of this optimal position is approximately equal to 0.038 (or a gap of 5 cm). To investigate the maximum resolution of the sensor, we adjusted the gap between two coils to nearly 4 cm and applied light weights to the free end of the cantilever beam; each weight can apply a strain of 0.6 με to the surface of the cantilever beam. As shown in Figure 9b, the reflection of the transmitter reached a maximum value of −69 dB when the strain is equal to 1.2 με. Due to the high sharpness of the signal, we can identify the resonant frequency shift of the signal at each loading step; this illustrates that the maximum resolution is better than 1 με.

The second experiment investigates the resolution of the sensor when the gap between coils was relatively large. The resolution is determined by the sensitivity, the quality factor, and the data noise of the sensor. Unlike conventional antennas, we prefer to improve the quality factor of the signal so that sharp peaks can indicate precise resonant frequencies. Besides the optimal position, the quality factor is almost constant for different wireless sensing distances (see Figure A5). Due to the interference of data noise, the resolution will decrease with the increase in the gap. When the gap increased from 5 cm to 10 cm, the resolution of the sensor was approximately 10–20 με (see Figure A6 in Appendix D). To enhance the resolution of the sensor, we increased the transmitter’s resistance *R_t_* to approximately 50 Ohms via connecting resistors in series with tuning capacitors. Figure 9d shows that the reflection increased to −31 dB (or 47.5 Ohms) when the gap was approximately 8 cm. The resolution was approximately 10 με, which was slightly higher than the resolution when no compensating resistor was used.

The third experiment shows that the passive wireless sensor can detect the variation of strain when the coupling coefficient is extremely low (see Figure 9c). We increased the separation distance up to 10 cm; the coupling coefficient *κ* is 0.0067. In each load step, we applied a weight of 1 kg to the cantilever beam, which is equivalent to a strain of 100 με. Although the magnitude of the receiver’s peak signal reduced significantly with the increase in wireless sensing distance, we can recognize the resonance shift with the strain variation. The wide peak signal reduced the resolution of the passive wireless sensor, so it can only be used to detect the change of large strains. The sensitivity is approximately 4 Hz/με. The resonant frequency increased non-linearly with the strain, due to the nonlinear relationship between the resonant frequency and the contact force.

The fourth experiment shows the influence of wireless sensing distance on the sensitivity of a BAW passive wireless sensor. When the separation distance was 5 cm, 7 cm, and 9 cm, the effect of different payloads on the signal were shown in Figure 10a–c, respectively; the signal-to-noise ratios were approximately 48 dB, 28 dB, and 15 dB, respectively. To process the peak signal with a low signal-to-noise ratio, we used curve fitting techniques with rational functions to filter the data noise and detect the frequency at the tip; both degrees of the numerator and denominator were five. Figure 10d,e shows the results of curve fitting when the separation distance was 5 cm and 9 cm, respectively; the former has a much higher signal-to-noise ratio than the latter. The resonance of the peak signal is the frequency at which the derivative of the reflection curve is approximately equal to 0. Thanks to the signal processing method, the longest sensing distance of the sensor was up to 12 cm (see Appendix D). We used this method to extract the resonant frequency of each dataset and calculate the resonance shift when the distances between coils were different (see Figure 10f). The results show that the resonance shift of the sensor was nearly constant if the sensing distance was longer than 5 cm. In this demonstration, we replaced the BAW strain sensor, so the sensitivity is approximately equal to 5.6 Hz/με, which is slightly higher than the sensitivity in Figure 9d. The difference in the sensitivity is due to the low repeatability of our manual assembly, which could result in the parallelism error between the Bridge and substrate of the BAW strain sensor.

The last experiment investigated the time response of the passive wireless sensor. As a reference, we glued a strain gauge on the other side of the cantilever beam (see Figure 11a). The strain gauge was connected with a strain gauge measurement module BF350-6AA and the output voltage was measured by an analog input of an Arduino Uno. We calibrated the strain gauge before the experiment. The gap between coils was 5.5 cm (*κ* = 0.04). To analyze the signals in real time, we used a portable vector network analyzer Mini-VNA PRO to collect the data and processed the raw data through our signal processing algorithm as described above (see Figure 11b). During the experiment, we applied multiple weights to the free end of the beam; each weight is 1 kg. As shown in Figure 11a, the time response of a BAW passive wireless sensor can basically match that of a commercial strain gauge, but we can see a few problems. First, the sensor has a slow refresh rate, approximately 2 Hz. The average time consumptions of the data measurement and the curve fitting were 0.28 s and 0.19 s, respectively. Second, the sensor shows that the resonant frequency of the sensor increased with time when the payloads were applied. This is due to the viscoelasticity of the rubber spacer. A steel spring may overcome this issue, but it could have a buckling issue.

## 4. Discussion

A comparison between this work and previous relevant work is shown in Table 4. The wireless sensing distance depends on the size of the coil, so we define a normalized distance to evaluate the relative wireless sensing distance of near-field passive wireless sensors, which is the spacing between coils divided by the coil diameter. Although our sensor has a modest sensitivity, our sensor has two advantages.

First, the sensor can detect the resonance of receiver when the coupling coefficient is extremely low. Due to the high quality factor of the quartz, the current in the transmitter has a recognizable change in a narrow frequency band when *κ* is low. In addition, the tuning capacitor in the transmitter increases the output power of the transmitter coil.

Second, the resonant frequency of the sensor is nearly independent of the wireless sensing distance if the coupling coefficient *κ* is low enough. One reason is that the difference in quality factors between the transmitter and the receiver is relatively higher than conventional inductor-capacitor (LC) based passive wireless sensors. Another reason is that the long sensing distance allows the sensor to measure the peak signal when *κ* is relatively low. In this case, the separation of eigenvalues caused by the variation of the coupling coefficient can be neglected. This property suggests that the sensor has a high stability when there is a misalignment or relative motion between coils.

The BAW passive wireless sensor may be compatible with a UAV-based monitoring platform for strain measurement of large buildings. The high stability of the sensor can reduce the measurement error due to the non-stationary motion or position error of the UAV. Thanks to the long wireless sensing distance, the UAV can collect the sensor signal at a safe distance. The operating frequency of the sensor is much lower than the communication bands of the UAV, so the signal interference to the sensor can be minimized. To mount the sensors, we propose a non-contact sensor-mounting method based on magnetic suction and pressure-curing adhesives. We will embed permanent magnets into the left and right supports of a sensor and spray pressure-cured adhesives on its substrate. The UAV will carry a sensor mounting module and approach the steel surface of the measured structure. The module includes a magazine-like structure that releases the sensor when the door is opened. The sensor will be attached to the measured surface by magnetic suction. The adhesive will cure under the pressure. This concept is shown in Figure 12.

Finally, we point out the problems of the prototype, which should be considered in future work. The first issue is the repeatability of the sensor. The manufacturing processing error, such as the parallelism errors between the contact unit and the wafer, will lead to a difference in sensitivity. The second issue is that the BAW sensor in this article is designed to measure the surface curvature of the measured material, so it is not sensitive to the tensile strain (see Figure A1b in Appendix B). The bridge-like structure converts the rotation between two supports into the normal force applied to the wafer surface. To measure the tensile strain, we can rotate the current structure by 90 degrees. In this case, the wafer is vertical to the measured material and sandwiched between two supports on the left and right sides. The tensile strain will separate two supports so that the contact pressure decreases.

The third issue is the influence of environmental conditions on BAW strain sensors, such as humidity and temperature. As shown in Figure 13a, the resonant frequency increases with increasing temperature. The thermal expansion of the sensor housing applied an added force to the wafer, which is superimposed on the force due to the deformation of the measured material. Therefore, the sensor requires temperature compensation to reduce the thermal effect on real applications. Figure 13b shows the compensated results when the room temperature was inconstant. According to the relationship between the force applied to the wafer and the resonance shift, we can calculate the force due to thermal expansion *F_temp_* and the total force applied to the wafer *F_total_*. Thus, the force due to strain follows *F_strain_* = *F_total_* − *F_temp_*. However, this method is unable to compensate for the effect of humidity. When water droplets are attached to the wafer surface, the droplets increase the mass of the wafer, so the resonant frequency decreases with increasing humidity (see Figure 13c,d). Unlike temperature, the added mass due to humidity and the added stiffness due to strain are not superimposable, so the calibration of humidity is beyond the scope of this paper. As an alternative solution, a sealed sensor package can separate the wafer from the moisture.

## 5. Conclusions

In this paper, we present the principle and design of a bulk acoustic wave strain sensor for passive wireless sensing. To the best of our knowledge, this is the first BAW sensor that can be used for strain measurement. The BAW sensor has a higher quality factor than a conventional LC-based strain sensor, so the resonant frequency of a BAW passive wireless sensor can be recognized when the coupling coefficient is relatively low. We established analytical and numerical models to investigate the contact mechanisms between the quartz wafer and the sensor housing. In addition, we used a prototype BAW passive wireless strain sensor to evaluate the sensor performance. The maximum resolution is 0.6 με, the sensitivity is constantly 5.6 Hz/με, and the maximum sensing distance is up to 10 cm.

## Figures and Tables

**Figure 1 sensors-23-03904-f001:**
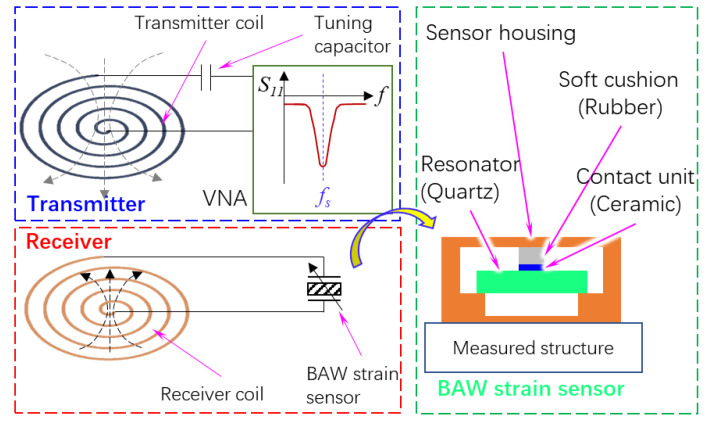
Schematic of a BAW passive wireless strain sensor.

**Figure 2 sensors-23-03904-f002:**
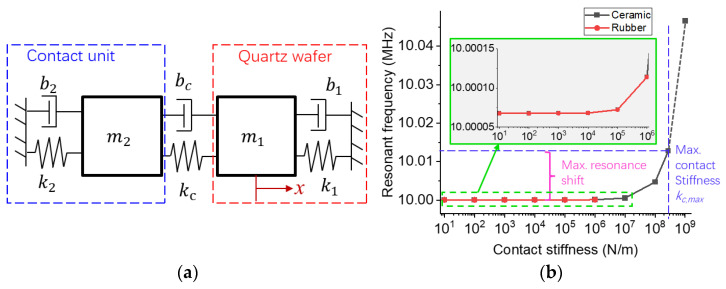
The effective model of a BAW strain sensor. (**a**) A dual mass-spring-damper model. (**b**) The influence of the effective contact stiffness on the resonance of the quartz.

**Figure 3 sensors-23-03904-f003:**
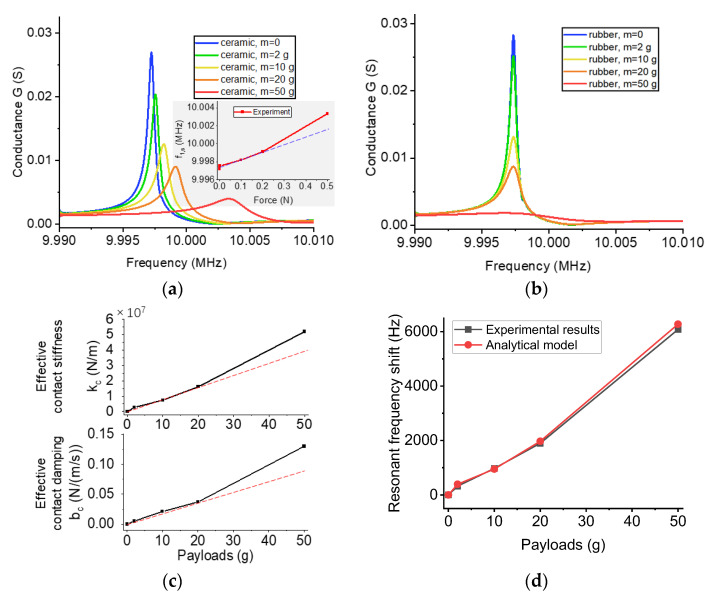
Experimental characterization of quartz wafers in contact with different contact units. (**a**,**b**) Conductance vs. resonance for a contact unit made of ceramic and rubber, respectively. (**c**) The effective contact spring and damping coefficient as a function of payloads. (**d**) Payloads vs. resonance shift in experimental and analytical results.

**Figure 4 sensors-23-03904-f004:**
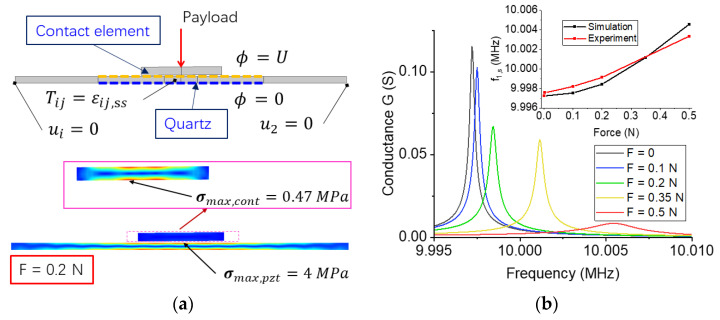
The numerical model of contact between a quartz wafer and a ceramic sheet. (**a**) The boundary conditions and model. (**b**) The conductance of wafer when different payloads are applied to the contact unit.

**Figure 5 sensors-23-03904-f005:**
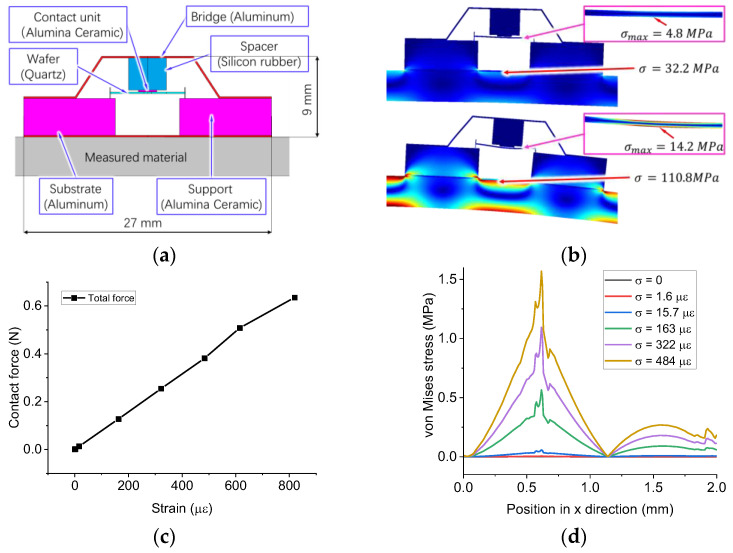
The numerical model of the sensing housing. (**a**) The design of the sensor. (**b**) The deformation and stress distribution of the sensor when the beam is bent into different shapes. The strain of the measured surface is 163 με and 484 με in the upper and lower case, respectively. (**c**) The total contact force as a function of the strain. (**d**) The von Mises stress distribution along the contact unit.

**Figure 6 sensors-23-03904-f006:**
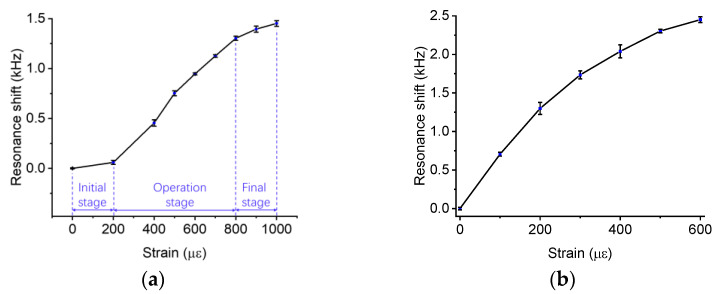
The calibration curves of two BAW sensors with different preloads. Each experiment is repeated five times. (**a**) The calibration curve of a sensor with a lower preload. (**b**) The calibration curve of a sensor with a higher preload. The average sensitivities of Sensor A and B are 1.4 Hz/με and 4.1 Hz/με, respectively. The maximum resonant frequency shifts are 1452 Hz and 2451 Hz, respectively. The maximum strain that these two sensors can measure is approximately 1000 με and 600 με.

**Figure 7 sensors-23-03904-f007:**
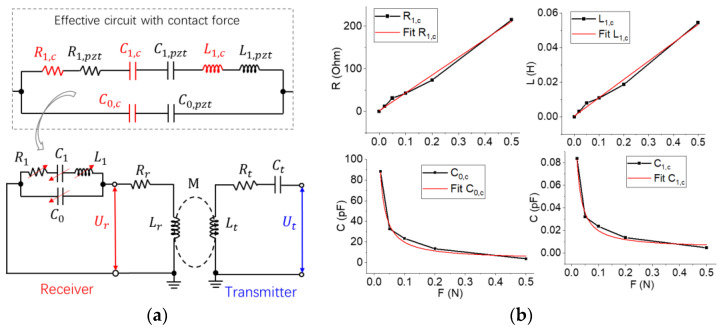
The electrical properties of a BAW passive wireless sensing system. (**a**) The circuit diagram of the system with lumped parameters. *R*_1,*c*_, *L*_1,*c*_, *C*_1,*c*_, and *C*_0,*c*_ are the added lumped parameters because of the contact. (**b**) The influence of compression on the four added terms.

**Figure 8 sensors-23-03904-f008:**
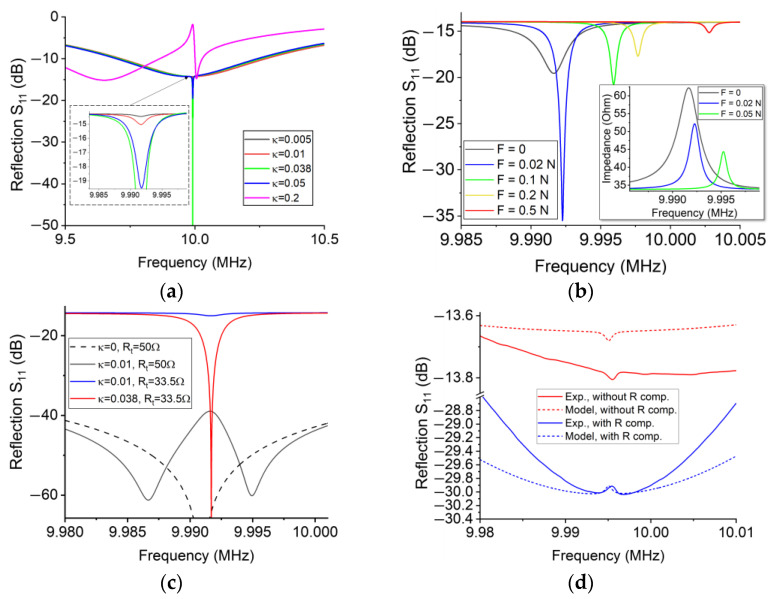
Characterization of the signal of BAW passive wireless sensors. (**a**) Coupling coefficient *κ* vs. reflection *S*_11_. No force is applied to the wafer. (**b**) Contact force vs. reflection *S*_11_. *κ* is 0.05. (**c**) Tunable optimal positions. The black solid curve is the optimal position for *κ* = 0, whereas the red solid curve is the optimal position for *κ* = 0.038. (**d**) Experimental verification of the compensation of the real part of the transmitter. According to the experimental measurement, we adjusted the coefficients of the model. The experiment result was the average of five repeated measurements.

**Figure 9 sensors-23-03904-f009:**
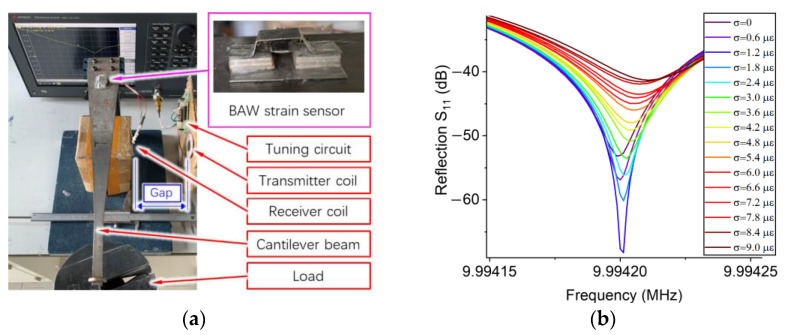
The design and demonstration of a prototype BAW passive wireless strain sensor. All signals are the average of five repeated measurements. (**a**) Experimental setup. (**b**) Sensor signals vs. strain at the optimal position. (**c**) Sensor signal vs. strain when gap was 8 cm. (**d**) The effect of strain on the sensor signal when the gap between coils was 10 cm.

**Figure 10 sensors-23-03904-f010:**
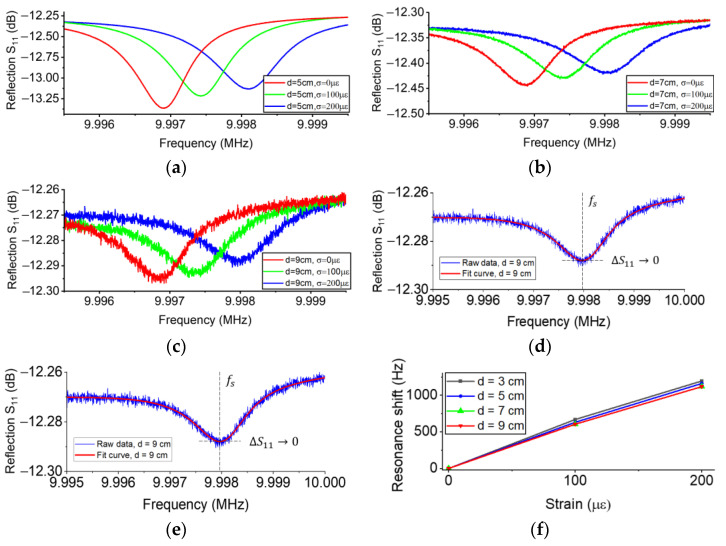
The effect of the wireless sensing distance on the sensitivity of a BAW passive wireless strain sensor. Resistance compensation was not used. All signals are the original data. (**a**–**c**) The effect of payloads on the signal when the distance was 5 cm, 7 cm, and 9 cm, respectively. (**d**,**e**) The curve fitting of the signal when the distance was 5 cm and 9 cm, respectively. (**f**) The resonance shift as a function of strain when the distance was different.

**Figure 11 sensors-23-03904-f011:**
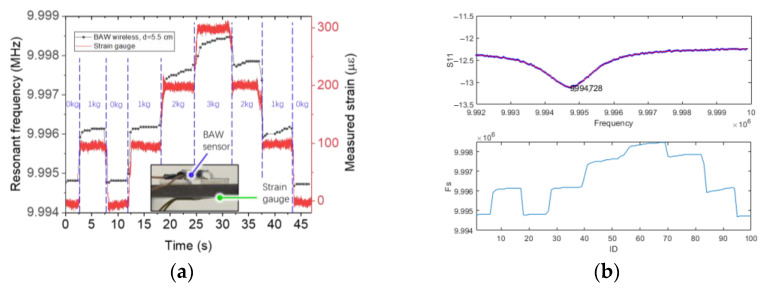
The time response of a BAW passive wireless sensor. (**a**) The signals of a BAW passive wireless sensor and a strain gauge as a function of time. (**b**) A screenshot of the real-time strain monitoring and signal processing interface. We measured 101 data points in 49 s.

**Figure 12 sensors-23-03904-f012:**
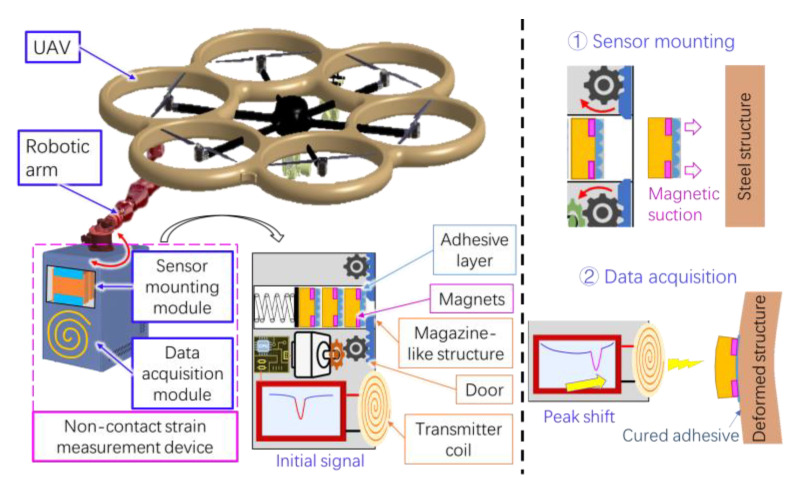
The concepts of non-contact sensor mounting and data acquisition. The strain measurement device is connected to a robotic arm integrated into a UAV. The module includes two modules: sensor mounting module and data acquisition module.

**Figure 13 sensors-23-03904-f013:**
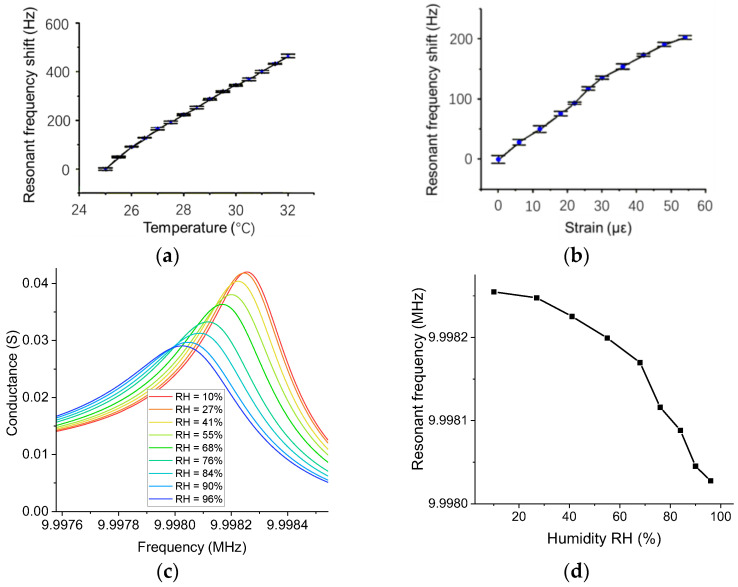
The effect of environmental conditions on a BAW passive wireless strain sensor. (**a**) Temperature vs. resonance shift. (**b**) The calibration of strain with temperature compensation. In (**a**,**b**), all experiments were repeated five times. (**c**) Humidity vs. conductance. (**d**) Humidity vs. resonance shift.

**Table 1 sensors-23-03904-t001:** The coefficients of an effective model for a BAW strain sensor.

Object	Dimensions (mm)	Shear Modulus (GPa)	Density (kg/m^3^)	Effective Mass *m_i_* (g)	Effective Spring Coefficient *k_i_* (N/m)	Contact Stiffness (N/m)
1 (Quartz)	5 × 5 × 0.161	27.4	2651	0.0106	4.21 × 10^10^	/
2A (Ceramic)	2 × 2 × 0.5	149	2300	0.0046	3.92 × 10^9^	1.51 × 10^8^
2B (Rubber)	5 × 5 × 2	0.018	1100	0.055	8.22 × 10^6^	2.06 × 10^5^

**Table 2 sensors-23-03904-t002:** The coefficients of an equivalent model for a BAW strain sensor.

Mass (g)	0	2	5	10	20	50
*R*_1_ (Ohm)	37.12	49.08	68.41	79.62	110.76	252.16
*L*_1_ (H)	0.0094	0.012	0.017	0.020	0.028	0.064
*C*_1_ (pF)	0.027	0.020	0.014	0.013	0.009	0.004
*C*_0_ (pF)	27.5	20.9	14.9	12.7	9.07	3.47

**Table 3 sensors-23-03904-t003:** The lumped parameters of a BAW passive wireless sensor.

Subsystem	Lumped Parameters	Value
Receiver	*R_r_*	10.2 Ω
	*L_r_*	6.3 μH
	BAW sensor	See Table 2
Transmitter	*R_t_*	33.5 Ω
	*L_t_*	6.3 μH
	*C_t_*	40.28 pF

**Table 4 sensors-23-03904-t004:** The comparison between this work and previous work.

Reference	Year	Sensitivity (Hz/με)	Test Distance *d* (mm)	Coil Diameter *Φ* (mm)	Normalized Distance *d*/*Φ*
[10]	2006	11	120	90	1.3
[23]	2014	4	5	45	0.1
[24]	2017	1.3	20	40	0.5
[25]	2021	100	2	15	0.1
[26]	2022	100	7	26	0.3
This work	2023	4	100	43	2.3

## Data Availability

The data and data processing methods are contained within the article, such as the curve fitting algorithm, and Appendix A.

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
