# Peer review of "A Bulk Acoustic Wave Strain Sensor for Near-Field Passive Wireless Sensing"

_sensors, 2023, doi:10.3390/s23083904_

Round 1
Reviewer 1 Report
The manuscript with the title “A Bulk Acoustic Wave Strain Sensor for Near-field Passive 2 Wireless Sensing” proposed a method of near-field passive wireless strain sensing. This paper is organized quite well. However, there are still some issues that need to be answered or solved by the authors before publication.
1 The size and materials parameters in figure 1 and 5a must be included.
2 The software and its version for simulations must be included.
3 Regarding figure 7b, why did the intensity (dB) of the frequency at different contact forces change a lot? Can we use the intensity (dB) for demodulation? How about the resolution if the authors use this way to process data? As I only saw the linear response, what is the maximum force that the sensor can be used? There should be a standard curve (https://en.wikipedia.org/wiki/Calibration_curve, including std) for the relationship between the strain force and frequency or intensity change.
4 The maximum resolution of the sensor with different gap distances between two coils can be plotted when the reflection of the transmitter reaches a maximum value. This will help readers better know the sensor’s performance.
5 Considering figure 9, the signal-to-noise ratio (SNR) corresponding to separation distance should be compared.
6 If a traditional humidity or temperature sensor is put inside the sensor, can this sensor calibrate or eliminate the humidity or temperature effect? How about the specificity? Please also discuss this part in the conclusion.
Reviewer 2 Report
In this paper, the principle and design of a bulk acoustic wave strain sensor for passive wireless sensing is presented. The paper is well written and deserve for publication. I have a few minor comments in order to improve the quality of the paper:
General comment line 35 - 37 - 40 and following: citations appear after the period, and this looks quite odd, as it seems the reference refers to the following sentence. All references must be editorially placed correctly.
If the supplemental documents will not be published, I wonder how the authors can include them in the paper. My suggestion is, for the suplementar material which are strictly required to put them in the text and for the others to remove them at all and do not cite them. Alternatively, they can be put in Appendices.
line 115 --> resistence of the receiver
line 174 we use an experimental method
line 179 --> top of the loading cell
line 251 --> more details about the model are shown
line 292 --> reflection of the transmitter
line 298 --> real part if the transmitter
line 307 --> resonance of the receiver increases
line 366 --> the resonance shift of the sensor was nearly constant
Author Response
Dear reviewer,
Thank you for your comments. We appreciate your help in improving the quality of this article. We have revised our articles according to your questions. Please check our response as follows.
- General comment line 35 - 37 - 40 and following: citations appear after the period, and this looks quite odd, as it seems the reference refers to the following sentence. All references must be editorially placed correctly.
Answer: Thank you for your reminder. We have modified the format of citations.
- If the supplemental documents will not be published, I wonder how the authors can include them in the paper. My suggestion is, for the suplementar material which are strictly required to put them in the text and for the others to remove them at all and do not cite them. Alternatively, they can be put in Appendices.
Answer: We apologize for missing these materials in the article. We have put all supplemental materials in Appendices.
- line 115 --> resistence of the receiver
line 174 we use an experimental method
line 179 --> top of the loading cell
line 251 --> more details about the model are shown
line 292 --> reflection of the transmitter
line 298 --> real part if the transmitter
line 307 --> resonance of the receiver increases
line 366 --> the resonance shift of the sensor was nearly constant
Answer: Thank you for your careful review. We have modified above issues in the article.
Reviewer 3 Report
The authors proposed an interesting idea for a wireless bulk acoustic wave strain sensor. The described details may be useful to readers for designing new sensors. I consider that the article is appropriate for publishing in Sensors after minor changes.
1. Unfortunately, I did not see any significant advantages of the presented sensor over classical sensors. Do the authors have any results comparing the data obtained from the new sensor and, for example, a classic strain gauge? Such information, in my opinion, should be presented.
2. What effect do torsional or multi-axial stresses have on sensor signals?
3. How is the sensor attached to the controlled object? What is the longest possible distance between coils?
